# Detection of Influenza D-Specific Antibodies in Bulk Tank Milk from Swedish Dairy Farms

**DOI:** 10.3390/v15040829

**Published:** 2023-03-24

**Authors:** Ignacio Alvarez, Sara Hägglund, Katarina Näslund, Axel Eriksson, Evelina Ahlgren, Anna Ohlson, Mariette F. Ducatez, Gilles Meyer, Jean-Francois Valarcher, Siamak Zohari

**Affiliations:** 1Division of Ruminant Medicine, Department of Clinical Sciences, Swedish University of Agriculture Sciences, 8 Almas Allé, 75007 Uppsala, Sweden; 2Department of Microbiology, National Veterinary Institute, Ulls väg 2B, 75189 Uppsala, Sweden; 3Växa Sverige AB, Uppsala, Ulls Väg 29A, 75651 Uppsala, Sweden; 4IHAP, Université de Toulouse, INRAE, ENVT, 31076 Toulouse, France

**Keywords:** Influenza D, cattle, serology, bovine respiratory disease, ELISA, bulk tank milk

## Abstract

Influenza D virus (IDV) has been detected in bovine respiratory disease (BRD) outbreaks, and experimental studies demonstrated this virus’s capacity to cause lesions in the respiratory tract. In addition, IDV-specific antibodies were detected in human sera, which indicated that this virus plays a potential zoonotic role. The present study aimed to extend our knowledge about the epidemiologic situation of IDV in Swedish dairy farms, using bulk tank milk (BTM) samples for the detection of IDV antibodies. A total of 461 and 338 BTM samples collected during 2019 and 2020, respectively, were analyzed with an in-house indirect ELISA. In total, 147 (32%) and 135 (40%) samples were IDV-antibody-positive in 2019 and 2020, respectively. Overall, 2/125 (2%), 11/157 (7%) and 269/517 (52%) of the samples were IDV-antibody-positive in the northern, middle and southern regions of Sweden. The highest proportion of positive samples was repeatedly detected in the south, in the county of Halland, which is one of the counties with the highest cattle density in the country. In order to understand the epidemiology of IDV, further research in different cattle populations and in humans is required.

## 1. Introduction

Bovine respiratory disease (BRD) is one of the most common and extensively studied diseases in the cattle industry worldwide, with a negative impact on production, animal welfare and health [1]. Several factors, such as host immunity, environmental conditions and a wide diversity of pathogens, interact and promote the onset of the disease [2]. Thanks to the extended use of molecular tools, Influenza D virus (IDV) was recently identified as a new virus involved in BRD [3,4]. This member of the Orthomyxoviridae family was first detected in pigs with respiratory symptoms, but subsequently to epidemiological analyses that revealed a high seroprevalence in cattle, bovidae was proposed to be the main reservoir [5].

Investigations based on the detection of both IDV-specific antibodies and virus demonstrated that IDV is present in cattle herds on several continents and in numerous cases with high infection rates [6,7,8,9,10,11,12]. The role of IDV in the pathogenesis of BRD is still unclear, but experimental infections of calves suggested that IDV alone can cause respiratory signs and lesions and that this virus has the ability to replicate in both the upper and lower respiratory tracts. However, IDV can also be detected in healthy animals, and the potential role of IDV in co-infections should be studied further [7,13,14,15].

To address the zoonotic role that IDV can represent, IDV was shown to replicate well in human airway epithelial cells, and seroepidemiological studies demonstrated IDV-specific antibodies in up to 95% of occupational workers in contact with cattle in USA, as well as in the general human population in Italy [16,17,18,19]. On the other hand, 3300 archived human respiratory samples collected in Scotland were tested for IDV by PCR, and no IDV-RNA was detected [20]. 

The enzyme-linked immunosorbent assay (ELISA) is well recognized as a robust technique with a good performance for the detection of antibodies against different pathogens in serum. Furthermore, the combination of ELISA with the use of bulk tank milk (BTM) samples provides a powerful tool to detect and monitor the prevalence of pathogens. However, there are potential risks of false negative results caused, for example, by a low seroprevalence in milked cows and the dilution effect caused by including a high number of seronegative animals in the BTM. Nevertheless, since the assay is simple, rapid and cheap, performing ELISA on BTM is considered an excellent option to detect and monitor the presence of infections in dairy cattle for surveillance and control programs’ purposes [21,22,23].

Based on the lack of information about the existence of IDV in Sweden and the potential importance that this pathogen might have for cattle production and human health, this study aimed to determine the presence and spatio-temporal distribution of IDV antibodies in the Swedish cattle population using an in-house indirect ELISA on bulk tank milk samples.

## 2. Materials and Methods

### 2.1. Sample Collection

During the spring of 2019 and 2020, 461 and 338 BTM samples, respectively, were collected following a risk-based design within the framework of the Swedish Surveillance Program for bovine viral diarrhea virus. The samples were collected from dairy herds across the country in a blinded manner and focused on counties with the highest cattle density. The counties were categorized into 3 regions: Norrland (North), Svealand (Central) and Götaland (South). The milk was collected from the bulk tank in vials containing bronopol as the preservative and transported to the National Veterinary Institute for further processing. At arrival, the vials were centrifuged at 3000 RPM to facilitate the removal of the cream fraction. Samples were stored at −20 °C until they were tested.

### 2.2. Indirect ELISA

An in-house indirect ELISA was performed to detect the presence of IDV-specific IgG antibodies, as described previously [15]. We previously demonstrated that the sensitivity and specificity of this test were 87% and 100%, respectively, when analyzing serum and using a haemagglutination inhibition assay as the gold standard [24]. The positive control used in the present study was an IDV-antibody-positive serum obtained from a calf that had been experimentally infected with IDVstrain D/bovine/France/5920/2014 (HI titer: 1/1024). This positive IDV control serum had been serially diluted tenfold in both IDV antibody-negative milk and sera from cattle and had been analyzed with both HI and ELISA. The dilution that resulted in an OD value 10% higher than the mean OD value of the negative sera, plus three standard deviations, was used as the positive control. Samples were considered IDV-antibody-positive if the corrected optical density (COD = OD_sample_ − OD_blank_) was >10% of the COD of the positive control. 

### 2.3. Map Design

Choropleth maps showing Sweden with the different counties and regions were created by using the software QGIS (3.26.0-Buenos Aires).

## 3. Results

Overall, 147 (32%) and 135 (40%) BTM samples collected in 2019 and 2020, respectively, were IDV-antibody-positive. In 2019, all the BTM samples from the region of Norrland were negative for IDV-specific antibodies; however, in 2020, the presence of two positive samples in the counties of Västerbotten and Jämtland within the region of Norrland allowed us to also confirm the presence of the virus in the most northern region of the country (Table 1). No IDV-specific antibodies were detected in Norrbotten (Norrland), Gävleborg (Norrland), Västernorrland (Norrland), Västmanland (Svealand), Stockholm (Svealand), Uppsala (Svealand) or Kronoberg (Götaland). Counties located in the southern region of Sweden (Götaland), and in particular in the county of Halland, had the highest proportion of antibody-positive samples (84% and 77% in 2019 and 2020, respectively, Table 1, Figure 1). Whereas the proportion of positive samples increased by 22% between 2019 and 2020 in the southern region, it increased by only 8.1% in the country. 

## 4. Discussion

These results demonstrate, for the first time, a circulation of IDV among Swedish dairy cattle. The southern region of the country (Götaland) consistently contained the highest number of antibody-positive herds, which suggests a high circulation of virus in that area. This was previously additionally demonstrated for bovine respiratory syncytial virus and bovine coronavirus and may be explained by the high density of animals [25]. Moreover, the shorter distance between farms may facilitate the transfer of animals between herds, more human contacts and animal professional visits in a higher number of herds per day. In addition, the connection with continental Europe might constitute an increased risk of transmission of infectious diseases. Halland, the county with the highest IDV-seroprevalence, has the highest density of dairy cows in Sweden (419 dairy cows per 100 km^2^) and is situated very close to the border with Denmark. In contrast, no IDV-specific antibodies were detected in samples from the county of Norrbotten, which has the lowest dairy cow density in Sweden (4.59 dairy cows per 100 km^2^) and which is situated at the opposite extreme of the border. Although not much information is available on the potential routes of transmission for IDV, experimental studies have shown that aerosol transmission within a building is possible [15]. Since IDV antibodies were detected in humans and in a wide range of animals, including wild animals such as wild boar, these may additionally contribute to the spread of the virus [26,27,28,29]. However, considering the fragility of influenza viruses in the environment, any human or animal species probably needs to be in the same building, and perhaps even in close contact with cattle, for the virus to spread.

Several serological studies conducted worldwide show a divergent seroprevalence in animals, but few of those studies report a seroprevalence at the herd level. Japan, Argentina and Luxembourg showed 50%, 75% and 97.7% IDV-antibody-positive farms among those sampled, respectively [8,12,24]. The broad sample design of this study, involving areas with a low and high animal density, and the high biosecurity level of the farms in Sweden may explain why the prevalence was not as high as in other countries. 

To our knowledge, this is the first IDV-specific antibody survey performed by using BTM. Although adult cows are commonly less susceptible to respiratory viral infections than calves, the fact that they are kept in herds for a long period of time turns them into good sentinel animals for this kind of surveillance. Bulk tank milk samples can be obtained in a fast, inexpensive and non-invasive way and are an excellent option for the mass screening of diseases, as long as no classic vaccine has been used, as in the case of IDV. On the other hand, although immunoglobulin G is the most abundant immunoglobulin found in milk, the antibody concentrations are considerably lower than those found in serum [30]. The dilution effect that is generated when samples from pooled milk are analyzed does not allow one to exclude with precision the presence of a positive animal. In addition, several authors point out that antibody levels in milk may fluctuate depending on different factors, such as the stage of lactation in which the animal is at the time of sampling, subclinical mastitis, the season, the feeding system, the breed, and the age [31,32,33]. Regarding IDV, no studies have yet established the time during which the antibodies can be detected in milk. Consequently, it is possible that the rates reported in this study are underestimated. Therefore, once the herd has been identified as positive by BTM samples, follow-up tests with individual serum samples can be performed to further identify animals exposed to IDV. In addition, IDV-RTqPCR can be routinely carried out on nasal swabs to improve detection and confirm or exclude viral circulation in herds in which the virus is circulating.

The observed shifts in the positive rate between 2019 and 2020 could be explained by differences in the sample size of each county for each year. However, if we consider the seasonality of influenza cases in humans, it is possible to hypothesize that this same pattern also occurs for Influenza D in the cattle population. Therefore, a variation in positive rates can be expected depending on the season and the year in which the sample is collected.

## 5. Conclusions

The present study demonstrates that IDV is present in Swedish dairy herds and with a higher prevalence in the south of the country. Further studies should focus on the detection of IDV in BRD outbreaks and on the duration of IgG antibodies in milk and serum. In addition, it is necessary to study the prevalence of IDV in other species including humans in order to gain a deeper understanding of the epidemiological pattern of this virus.

## Figures and Tables

**Figure 1 viruses-15-00829-f001:**
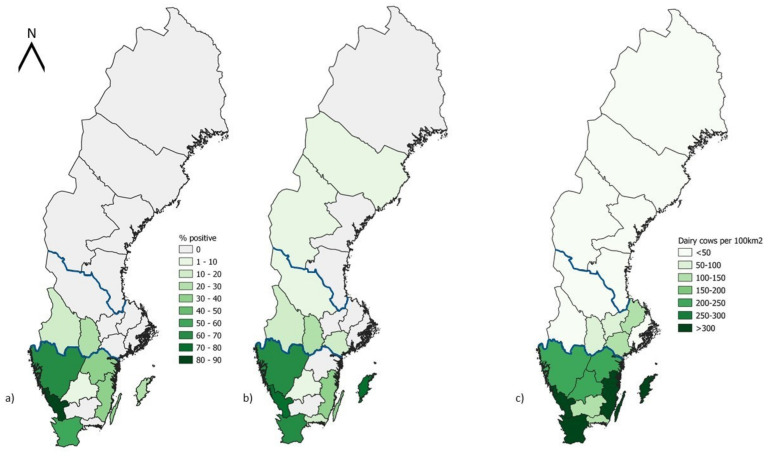
Maps of Sweden displaying the counties and the borders of the northern, central and southern regions. (**a**) The proportion of bulk milk samples in which IDV-specific IgG antibodies were detected in 2019, (**b**) the proportion of bulk milk samples in which IDV-specific IgG antibodies were detected in 2020, and (**c**) dairy cow density. The blue lines highlight the border of the different regions.

**Table 1 viruses-15-00829-t001:** Sample size, number and proportion of herds with IDV-specific antibody in bulk tank milk in different Swedish counties.

		2019			2020			Overall	
Region	County	Sample Size		NPS ^a^ (%) ^b^	Sample Size		NPS ^a^ (%) ^b^	Sample Size	NPS ^a^ (%) ^b^
Norrland (North)	Gävleborg	13		0 (0)	7		0 (0)	20	0
	Norrbotten	5		0 (0)	6		0 (0)	11	0
	Västerbotten	14		0 (0)	41		1 (2.4)	55	1 (1.8)
	Västernorrland	14		0 (0)	6		0 (0)	20	0
	Jämtland	5		0 (0)	14		1 (7.1)	19	1 (5.3)
Total Norrland		51		0 (0)	74		2 (2.7)	125	2 (1.6)
Svealand (Center)	Örebro	8		2 (25)	13		3 (23.1)	21	5 (23.8)
	Västmanland	10		0 (0)	3		0 (0)	13	0
	Dalarna	14		0 (0)	12		1 (8.3)	26	1 (3.8)
	Värmland	16		2 (13)	10		1 (10)	26	3 (11.5)
	Stockholm	5		0 (0)	7		(0)	12	0
	Uppsala	13		0 (0)	12		(0)	25	0
	Södermanland	19		0 (0)	15		2 (13.3)	34	2 (5.9)
Total Svealand		85	4 (4.7)	72	7 (9.7)		157	11(7)	
Götaland (South)	Östergötland	16		6 (38)	6		0 (0)	22	6 (27.3)
	Västra Götaland	76		51 (67)	57		39 (68.4)	133	90 (67.7)
	Gotaland	24		5 (21)	7		5 (71.4)	31	10 (32.2)
	Halland	37		31 (84)	13		10 (76.9)	50	41 (82)
	Blekinge	5		0 (0)	7		1 (14.3)	12	1 (8.3)
	Skåne	56		33 (59)	40		25 (62.5)	96	58 (60.4)
	Kalmar	47		15 (32)	61		46 (75.4)	108	61 (56.5)
	Jönköping	40		2 (5)	1		0 (0)	41	2 (4.9)
	Kronoberg	24		0 (0)	0		0	24	0
Total Götaland		325		143 (44)	192	126 (65.6)	517	269 (52)	
Total Sweden		461		147 (31.8)	338		135 (39.9)	799	282 (35.2)

^a^ Number of positive samples ^b^ Percentage of positive samples.

## Data Availability

Not applicable.

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
