# Peer review of "Detection of Influenza D-Specific Antibodies in Bulk Tank Milk from Swedish Dairy Farms"

_viruses, 2023, doi:10.3390/v15040829_

Round 1

Reviewer 1 Report

In the present study by Alvarez et al., the circulation of influenza D virus among Swedish dairy cattle was suggested by detecting IDV-specific antibody in their bulk tank milk collected from 21 districts of Sweden during 2019 and 2020. 

If the present study is the first antibody survey Influenza D Virus using bulk tank milk, I would like more information on the method. 

1. To apply milk samples to ELISA, how was the bulk tank milk treated? Was whey collected by centrifuging milk?

2. It is unclear why a sample showing OD value of >10% of the positive control was considered as positive. Thus, detailed information about "positive control" serum is necessary. For example, what absorbance does it show? 

3. Can sensitivity and specificity of this ELISA when using serum be applied to milk? 

4. Audiences will also want to know if the herds (farms) sampled in 2019 and 2020 overlap.

In Results section, the authors state that Vasterbotten is positive (line 72). However, the authors also state that No IDV-specific antibodies were detected in Vasterbotten (line 75). And Table 1 shows 1/55 (1.8%) of Vasterbotten as positive. Which is right?

Does the Choropleth map in Figure 1 show the county? And where are North, Center and South separated?

“Author's response” was cited as reference 6, but shouldn't you cite the original paper?

Author Response

REVIEWER 1

  1. To apply milk samples to ELISA, how was the bulk tank milk treated? Was whey collected by centrifuging milk?

The way in which the milk was collected, processed and stored before performing the ELISA has been added to the text. (line 74-77)

  1. It is unclear why a sample showing OD value of >10% of the positive control was considered as positive. Thus, detailed information about "positive control" serum is necessary. For example, what absorbance does it show? 

We have clarified this aspect according to the reviewer’s suggestion. (line 80-91)

  1. Can sensitivity and specificity of this ELISA when using serum be applied to milk?

Our intention was not to state that the sensitivity and specificity of the ELISA was the same for milk as for serum, but to highlight that ELISA has demonstrated a satisfactory diagnostic performance (evaluated within EFSSA project, C. Chiapponi, M. Ducatez, S. Faccini, E. Foni, M. Gaudino, S. Hägglund, A. Luppi, G. Meyer, A. Moreno, K. Näslund, N. Nemanichvili, J. Oliva, A. Prosperi, C. Rosignoli, V. Renault, C. Saegerman, A. Sausy, C. Snoeck, J-F. Valarcher, H. Verheije, S. Zohari. (2020) Risk assessment for influenza D in Europe. European Food Safety Authority (EFSA) supporting publication, ESR ; Volume17, Issue6 04 June 2020, https://doi.org/10.2903/sp.efsa.2020.EN-1853 ). To make it clearer for the reader, the sentence has been reformulated. (line 80-83)

  1. Audiences will also want to know if the herds (farms) sampled in 2019 and 2020 overlap.

In the frame of the BVDV surveillance program, the selection of the farms was blinded following a risk-based design; therefore we do not have access to the farm identification number and could not share this information with the reader. This aspect have been updated in material and method. (line 71-72)

  1. In Results section, the authors state that Vasterbotten is positive (line 72). However, the authors also state that No IDV-specific antibodies were detected in Vasterbotten (line 75). And Table 1 shows 1/55 (1.8%) of Vasterbotten as positive. Which is right?

In the text we state,

“Overall, 147 (32%) and 135 (40%) BTM samples collected in 2019 and 2020, respectively, were IDV-antibody positive. In 2019, all the BTM samples from the region of Norrland, were negative for IDV-specific antibodies, however in 2020, the presence of two positive samples in the counties of Västerbotten and Jämtland within Norrland allowed to confirm the presence of the virus also in the most northern region of the country.”

Our statements are correct since in 2020, we detected one IDV-antibody positive farm in each county. (in 2020, 1/55 and 1/19 of the samples collected in Vasterbotten and Jamtland, respectively, were IDV-antibody positive). We have clarified this statement in the manuscript (line 143).

  1. Does the Choropleth map in Figure 1 show the county? And where are North, Center and South separated?

Yes, it shows the counties and this is now specified in the legend Fig 1 and in the manuscript (line 156)  Following the question of the reviewer, we have outlined what is considered as North, Center and South appear on the maps (Fig 1). Changes in maps and legends has been made.

  1. “Author's response” was cited as reference 6, but shouldn't you cite the original paper?

We have corrected the reference citation.  

Reviewer 2 Report

This brief report by Alvarez et al. detects Influenza D virus antibodies in bulk tank milk from Swedish dairy farms via indirect ELISA. This prevalence reported is divided by northern, middle, and souther regions of Sweden. Considering a potential zoonotic transmission capability of IDV, this research is significant.

I believe that the materials and methods can be expanded. In the discussion section, the potential reasons behind the findings are well explained. Additional data on human visits to cattle at different areas, may have been informative. Overall, this study reveals novel and interesting findings which raises many more questions, like, can IDV antibodies in milk protect humans consuming them? How long are those antibodies stable?

Line 20: 269/517 is 52% and not 65%.

Line 25: "Bulk Tank Milk" is mentioned throughout the article, hence the last keyword must be changed to "Bulk Tank Milk".

Line 55: A subtitle "2.1 Sample collection" can be added before staring this paragraph.

Line 60: A subtitle "2.2 Indirect ELISA" can be added before staring this paragraph.

Line 99 and 101: square (2) should be in superscript after "km".

Line 130: Kindly suggest future solutions of this problem- maybe conducting qPCR or titration assays with virus in cattle tissues will provide a more correct estimate?

Line 134-135: Again, kindly suggest future solutions. If possible, add when the samples mentioned in Table 1 were collected each year?

Author Response

  1. Line 20: 269/517 is 52% and not 65%.

Thank you for pointing this out. The modification was made it in the text. (line 20)

  1. Line 25: "Bulk Tank Milk" is mentioned throughout the article, hence the last keyword must be changed to "Bulk Tank Milk".
  2. Line 55: A subtitle "2.1 Sample collection" can be added before staring this paragraph.
  3. Line 60: A subtitle "2.2 Indirect ELISA" can be added before starting this paragraph.
  4. Line 99 and 101: square (2) should be in superscript after "km".

Thank you for these suggestions. They have been integrated in the manuscript. Keywords, subtitles and square in superscript has been added to the text.

  1. Line 130: Kindly suggest future solutions of this problem- maybe conducting qPCR or titration assays with virus in cattle tissues will provide a more correct estimate?

As suggested by the reviewer, we have included a statement in the manuscript (line 206-210)

  1. Line 134-135: Again, kindly suggest future solutions. If possible, add when the samples mentioned in Table 1 were collected each year?

Bulk tank milk samples were collected during the spring in 2019 and 2020. This point has been added in the manuscript. (line 69)

Round 2

Reviewer 1 Report

- Materials and methods: I generally agree, but I don't understand what “corrected optical density (COD)” means. (Line 90 in the revised manuscript)

- Results: I pointed out that the statement that no antibodies were detected in Vasterbotten (line 134 in the revised manuscript) may be incorrect. I agree that no antibodies were detected in Vasternorrland. Please read the manuscript carefully. And I cannot find any sentence in line 143.

- References: Nothing has changed about reference 6 (line 245-247 in the revised manuscript), what did the author modify? 

Author Response

Dear 

In order to differentiate between new and old changes in the text, you will find the most recent modifications highlighted in yellow. 

Materials and methods: I generally agree, but I don't understand what “corrected optical density (COD)” means. (Line 90 in the revised manuscript)

-We have clarified this aspect in the manuscript (lines 91-92). Correct optical density is defined as “COD = OD sample – OD blank)”

Results: I pointed out that the statement that no antibodies were detected in Vasterbotten (line 134 in the revised manuscript) may be incorrect. I agree that no antibodies were detected in Vasternorrland. Please read the manuscript carefully. 

-We apology for this omission. The modification has now been made in the text (line 120)

References: Nothing has changed about reference 6 (line 245-247 in the revised manuscript), what did the author modify? 

-We apology for this.  We misunderstood your comment at your first review. The correct reference has now been added. (lines 252-253)